# The Effectiveness of Attachment Security Priming in Improving Positive Affect and Reducing Negative Affect: A Systematic Review

**DOI:** 10.3390/ijerph17030968

**Published:** 2020-02-04

**Authors:** Angela C. Rowe, Emily R. Gold, Katherine B. Carnelley

**Affiliations:** 1School of Psychology, University of Bristol, Bristol BS8 1TU, UK; 2Department of Psychology, University of Southampton, Southampton SO17 1BJ, UK; E.Gold@soton.ac.uk (E.R.G.); K.Carnelley@soton.ac.uk (K.B.C.)

**Keywords:** attachment, security, security priming, depression, anxiety, positive affect, negative affect

## Abstract

Attachment security priming has been extensively used in relationship research to explore the contents of mental models of attachment and examine the benefits derived from enhancing security. This systematic review explores the effectiveness of attachment security priming in improving positive affect and reducing negative affect in adults and children. The review searched four electronic databases for peer-reviewed journal articles. Thirty empirical studies met our inclusion criteria, including 28 adult and 2 child and adolescent samples. The findings show that attachment security priming improved positive affect and reduced negative affect relative to control primes. Supraliminal and subliminal primes were equally effective in enhancing security in one-shot prime studies (we only reviewed repeated priming studies using supraliminal primes so could not compare prime types in these). Global attachment style moderated the primed style in approximately half of the studies. Importantly, repeated priming studies showed a cumulative positive effect of security priming over time. We conclude that repeated priming study designs may be the most effective. More research is needed that explores the use of attachment security priming as a possible intervention to improve emotional wellbeing, in particular for adolescents and children.

## 1. Introduction

This systematic review evaluates the results and quality of studies using attachment security priming to reduce negative affect and/or improve positive affect in adults and children. Attachment styles represent individuals’ internalised histories of received care [1]. Experiences of care or rejection are abstracted to form trait-like mental models that drive style-congruent thinking, feeling, and behaviour [2,3]. Attachment styles are conceptualised along two dimensions: *anxiety regarding abandonment* and *avoidance of intimacy* [4]. Individuals can be low or high on either. Being high on either dimension is referred to in shorthand as being attachment ‘insecure’. Attachment styles are important predictors of the way individuals regulate affect. Individuals learn through relationships how and when to attend to their own stress and distress [5]. Each attachment dimension is associated with distinct affect regulation strategies [6,7], with insecure individuals experiencing lower positive affect and greater levels of negative affect relative to secures [8,9].

Individuals who are high in attachment anxiety find it hard to regulate their emotions. They use hyperactivation emotion regulation strategies, that is, they are hypervigilant for signs of rejection and have relationships characterised by emotional turbulence. Individuals high in attachment avoidance use deactivating emotion regulation strategies, that is, they ignore or deny emotional threats and tend toward compulsive self-reliance. By contrast, secure individuals (low on both dimensions) show optimal emotional regulation. While security of attachment has a number of positive wellbeing-related correlates and can be thought of as a protective factor against ill mental health, insecurity acts as a vulnerability factor for the onset of a wide variety of psychopathologies such as depression and anxiety [5].

In addition to a ‘global’ (or trait) attachment style, adults have relationship-specific styles based on their different long-term relationships; attachment styles are hierarchically organised [10]. A person’s global style is the most cognitively available and accessible style at the top of the hierarchy, while relationship-specific attachment styles (to parents, siblings, etc.) are lower down the hierarchy [11]. Relationship-specific attachment styles can be reliably primed supraliminally or subliminally. Once primed, they drive information processing, feeling, and behaviour, similarly to global attachment style. Supraliminal priming techniques include visualisation, while subliminal techniques expose participants to security-related stimuli below conscious threshold [12].

### Attachment Security Priming and Affect

Security priming has many positive personal and interpersonal effects, including the increase of self-esteem [13], prosocial values [14] and compassion and altruism [15]. It also reduces negative affect and increases positive affect [16,17,18].

We systematically reviewed the effectiveness of security priming compared to experimental and passive controls (no prime) in improving affect and considered the priming techniques and methods shown to be successful. We conceptualise affect as two separate constructs (positive vs. negative), reflecting the dominant approach in the literature and the one taken by the studies reviewed (defining our criteria for affect ensures the current systematic review is consistent, reliable and valid. Use of multiple models of affect [19] would compromise critical evaluation and comparisons between studies). Positive affect depicts feelings of pleasurable engagement with the environment such as happiness, excitement, and contentment [20]. Negative affect is defined as feelings of distress and unpleasurable engagement [21].

## 2. Method

### 2.1. Literature Search

Four electronic databases were used: PsychINFO, Web of Science, MEDLINE and Scopus (see Figure 1). An initial search between July 2018 to August 2018 yielded 355 results (it should be noted that some of the results were duplicates so the number of unique papers was smaller.) (101 PsychINFO, 66 Web of Science, 132 MEDLINE, 56 Scopus). Backward and forward chaining techniques, including examining reference lists and citations in the three key studies most closely related to our systematic review question [22,23,24] yielded 196 papers, resulting in 507 results in total. Searches were conducted using the following terms (a) attachment security priming, (b) affect, and (c) children and adults. The search terms for attachment security priming included prime*, priming and attach* or secur*. They also included affect* or effect* (using the word ‘effect’ for affect was to allow for spelling mistakes that may have occurred.) or anxi* or depress* or positive mood or negative mood, child*, young person, adolescen*, teen*, young adult*, adult*.

### 2.2. Screening Process

We used a two-step screening process to select the final papers. Step 1: Abstract and Method sections were screened for relevance. Step 2: Full texts were screened to determine whether the studies included attachment security priming to influence affect in the defined populations.

### 2.3. Inclusion and Exclusion Criteria

Inclusion criteria specified that the articles be published in English and in peer-reviewed journals. Articles were excluded if they used animals, were unpublished or were reviewed. Further papers were excluded if they did not use priming to influence affect, were duplications from other databases, were unsuitable populations, or if a full text was not available. This resulted in 469 papers being removed. Of the remaining 38 papers, qualitative studies were excluded (23) leaving a final set of 15 papers and 30 studies (see Table 1).

### 2.4. Structure and Framework of Review

Eligible studies were quality assessed for strengths and weaknesses using an established checklist [34]. The framework consists of 27 questions, sub-divided into five categories: reporting, external validity, internal validity bias, internal validity (selection bias), and power.

## 3. Results

### 3.1. Research Methodology

Participants. Across studies 3459 participants received priming. The samples largely consisted of university students, but two tested child and adolescent participants [24], [26] (b), one a clinical sample [27], and one heterosexual romantic couples [29] (c). Most studies reported age ranges between 6 to 76 year, mean—21.9 years but not all [25], [29] (a–c), [30]. Studies were conducted in five countries: twelve in Israel [17] (a–e), [33] (a–g), ten in the US [24], [26] (a,b), [29] (a–c), [30,31], [32] (b), six in the UK [13,18], [22] (a,b), [25,27], one in China [16] and one in Canada [32] (a). The average percentage of females across studies was 64.3%. Studies inconsistently reported socio-demographic status and ethnicity.

Research design. Seventeen studies utilised a between-subject experimental design [13], [17] (a–e), [18], [22] (a,b), [24,25], [26] (b), [27,28,30], [32] (a,b) and five used a within-subject experimental design [29] (c), [31], [33] (b-d). Seven studies used mixed method designs [26] (a), [29] (a,b), [33] (a,e–g), and one adopted a quasi-experimental design [6].

Measures. A range of measures were used. All the studies utilised at least one self-report measure; the most popular was the Experiences in Close Relationships (ECR) [4], used in nineteen studies [13], [17] (a–e), [18], [22] (a,b), [25], [26] (a,b), [28], [29] (a–c), [30], [31]. Global attachment dimensions were used as either independent variables (IVs), dependent variables (DVs) or covariates. Alternative attachment measures were used, including the attachment story completion task [35,36] and a 10-item measure of attachment anxiety and avoidance [37,38] based on established measures [4]. The most common measures of affect were Profile of Mood States [39] in some studies [22] (a,b), [27] and the Implicit Positive and Negative Affect Test [40] in others [25], [29] (b,c). Additional measures included interpersonal experience or expectation [13], [17] (a–e), [18], [25], [28], felt security [41] in a few studies [13], [22] (a,b), [27] and attachment figure information in others [18,27], [33] (d). Two studies used heart rate monitoring, facial expression coding [24], and functional magnetic resonance imaging (fMRI) brain scans [31].

Effect sizes. Effect sizes for group differences were reported in 23 studies. The type of effect sizes used included partial eta squared (*η*p^2^), eta-squared (*η*^2^), Cohen’s *f*^2^ and Cohen’s *d* [42]. Table 1 shows reported effects sizes, which have been classified as small, medium or large (following guidelines by [43,44].

### 3.2. Results by Type of Prime: Subliminal Primes

Sixteen studies employed a subliminal priming technique [17] (b–d), [24], [26] (a), [28,31], [32] (a,b), [33] (a–g). Of these, eleven used picture primes [17] (b,d), [24], [32] (a,b), [33] (a–c, e–g) and six used word primes [17] (c), [26] (a), [28,31], [33] (d,e).

Picture primes. Attachment security primes were pictures of a mother comforting her baby [24], [32] (a,b), [33] (a–c, e–g), a young heterosexual couple embracing [17] (b–d), [32] (a,b) and an elderly couple sitting close together [33] (c). Control primes (used in all 11 studies) varied, all used neutral (non-attachment relevant) pictures as a control, eight also used positive affect picture primes unrelated to attachment as controls, such as a beautiful natural scene [17] (b,d), [32] (b), [33] b,c, e–g), and four included a no picture (passive) condition [33] (a, e–g). Across studies, attachment security priming significantly reduced negative affect or increased positive affect in comparison to neutral or no picture controls. The effectiveness of positive affect priming relative to security priming varied; three studies found that positive affect primes [17] (b,d), [24] were less effective than attachment security primes, while three reported comparable results for the two primes [33] (a–c). Four studies manipulated participant stress and in these attachment security priming was more effective than positive affect priming [32] (b), [33] (e–g). Effect sizes ranged from medium [33] (a) to large [24], [33] (b,c, e–g).

Word primes. Six studies employed word priming tasks, including rating the similarity or liking of stimuli [26] (a), [28,31], [33] (d,e) and a lexical decision task [17] (c). Word primes included: common words related to attachment security versus neutral words [17] (c), [26] (a), [28,31], and names of the participants’ self-reported attachment figures versus names of close persons (but not attachment figures), associates and unknown individuals [33] (d). Three studies found attachment security word priming to be effective in reducing negative affect or increasing positive affect [17] (c), [28], [33] (d). Participants primed with attachment secure words reported significantly higher liking ratings for neutral stimuli [33] (d) and showed reduced maladaptive pain responses [28] compared to control words. They also showed reduced personal distress compared to a positive affect prime [17] (c). One study reported no differences between subliminally presented security word primes versus insecurity and neutral word primes in terms of participant responses to a prime manipulation check involving liking ratings of images [31] and another found that security word priming led to lower depressive symptoms when re-assessed one week later [26] (a). It should be noted however that the authors provided data on their combined subliminal and supraliminal findings, thus it was not possible to determine the distinct contribution of the subliminal prime alone.

Interaction with global attachment style. Eleven subliminal prime studies reported that global attachment style did not moderate the effects of prime [24], [26] (a), [28,31], [33] (a–g). Three studies, all designed to elicit empathy in the participant, reported that attachment anxiety moderated the effects of primed style [17] (b–d) but did not report effect sizes. Thus, for individuals high in global attachment anxiety, feelings of empathy rendered the secure prime less effective, maybe because empathy activated their global style which over-rode the prime.

### 3.3. Results by Type of Prime: Supraliminal Primes

Eighteen studies used a supraliminal priming technique [13,16], [17] (a,b,e), [18], [22] (a,b), [25], [26] (a,b), [27], [29] (a–c), [30,31], [33] (a).

Mental imagery task. Fourteen studies used mental imagery tasks [13,16], [17] (a,b,e), [18], [22] (a,b), [25], [26] (a,b), [27], [29] (a), [30]. Of these, 11 provided participants with a description of a secure attachment figure and asked them to think and/or write about the relationship/individual [13,16,18], [22] (a,b), [25], [26] (a,b), [27], [29] (a), [30] and 3 described a problematic interpersonal experience and asked participants to imagine they were in this situation and helped by an attachment figure [13], [17] (a,b). Eleven of the 14 studies found that security priming reduced negative affect or increased positive affect, relative to control [13], [17] (a,b), [18], [22] (a,b), [25], [26] (a,b), [27], [29] (a) and 3 studies reported no differences between the secure and neutral prime conditions in terms of anxious and depressed mood [22] (a), depressed mood [22] (b) and emotional wellbeing [30]. It should be noted that of the 11 studies in which the prime was effective, one did not reveal a statistical difference between the prime conditions [26] (b), one reported significant differences between prime conditions at one time point only [27] and two reported significant differences between prime conditions for one outcome variable only [29] (a), [30]. Based on seven studies, effect sizes ranged from medium [22] (a), [25,27] to large [13,16], [22] (b), [29] (a).

Picture or word task. Five studies used supraliminal pictures or word primes [26] (b), [29] (b,c), [31], [33] (a). All bar one [33] (a) reported reduced negative affect or increased positive affect in the secure prime condition relative to the control condition. Two studies used photographs of attachment figures, such as participants’ mothers [29] (b) or romantic partners [29] (c), two used picture or word primes related to attachment security [31], [33] (a). One study used a variety of security priming tasks: a picture writing task that required participants to describe a picture of a mother and baby, a study and recall of secure-themed sentences task, a secure word search task, and two visualisation/writing tasks in which participants wrote for 2 and 5 min about their secure experiences [26] (b). Interestingly, this study found that participants’ post-prime self-reported security differed as a function of prime type [26] (b), with the two visualisation tasks evoking higher security than the picture writing task. This suggests that mental imagery tasks in which participants are required to process information about their own attachment figures and/or attachment experiences, may be more effective than tasks that require the processing of information about an unknown other’s attachment secure interactions or experiences. Though it also should be noted that the participants in this study were teenagers (aged between 13–19 years) who may have found it challenging to relate to the experiences of mothers with babies. We cannot, therefore, rule out the possibility that the picture writing task may be more effective in older samples. Based on three studies that reported effect sizes, group differences between secure and control conditions were large [29] (b,c), [31].

Insecurity priming. Five supraliminal priming studies also primed attachment insecurity [17] (e), [18], [22] (a), [30,31]. Three found that primed insecurity led to greater negative affect than primed security [17] (e), [18], [22] (a), although primed avoidance did not result in greater depressed mood [22] (a) or higher personal distress [17] (e), compared to primed security. Furthermore, one study reported that priming anxious attachment led to small improvements in the participants’ positive affect over time, comparable to security priming [30]. None of the studies reported effect sizes.

Repeated priming. Five studies used repeated priming methodologies [13], [22] (b), [26] (b), [27,30]. One shot priming techniques have produced relatively short-lived effects [18,35]. In all five studies, the first subsequent prime was delivered 24 h after the initial prime. Two studies carried out an initial prime in the laboratory and then subsequent primes were delivered to participants via text message [22] (b), [27]. The remaining studies administered their initial and subsequent primes in the laboratory [13], in the participants’ naturalistic setting [26] (b), and/or via a study website [30]. Primes varied in length ranging from 3-min [22] (b), [27] to 10-min visualisations [13], [26] (b). Three of the studies kept their repeated primes constant and two used different primes [13], [26] (b). Studies generally reported that repeated security priming was effective in maintaining attachment security elevated over time [13,30]. One study showed a non-linear pattern to the effects of repeated security priming (e.g., lower state attachment security after the day 3 prime compared to the day 2 prime), possibly due to the differing efficacy of the different priming tasks used [26] (b).

Five studies measured the longer-term effects of security priming by administering post-prime dependent measures days, weeks or a month after the last prime [22], [26] (a), [27], [29] (c), [30]. Two studies collected dependent measures after each prime and at 24 h after the last security prime [22] (b), [27] and report that the secure prime resulted in higher felt security [22] (b), [27] and reduced anxiety and depression [22] (b), [27] compared to the neutral prime, at each time point. In both studies, however, reported security in the security prime group was lower at the final measurement relative to the previous time points. In other words, the positive effects of the secure prime appeared to decrease once participants were no longer exposed to it. Two studies collected post-prime dependent measures one week or more after the last prime was delivered [26] (a), [29] (c). One of these collected data one week after the last prime and found that self-reported depressed mood was significantly lower than at baseline for both secure and neutral priming conditions, although the decrease in the secure group was twice as large [26] (a). Another study measured post-prime emotional and physical health one month after the last prime and found that individuals primed with romantic partner (assumed to represent a secure attachment figure) experienced greater recovery in negative affect and continued to improve one month later [29] (c) (e.g., less physical pain and anxiety). Finally, a longitudinal study, involving priming attachment anxiety, security or a control every week for four months [30], showed reduced attachment anxiety over time in both the secure and anxious prime groups compared to the control group, while wellbeing and attachment avoidance were unaffected over time by prime. The security priming group also reported higher positive affect throughout the study than other control groups from the first measurement onwards.

Interaction with global attachment style. Fifteen supraliminal priming studies measured the interaction between primed and global attachment styles [13], [17] (a,b,e), [18,25], [26] (a,b), [27], [29] (a–c), [30,31], [33] (a). Eight reported no moderating effects of global style [13,18], [26] (a,b), [27,30,31,32], [33] (a) and two thirds of these used repeated priming designs. Three studies reported that global anxious style moderated the effects of primed style [17] (a,b,e). Congruent with the moderating effects of attachment anxiety in the subliminally presented security prime studies reviewed above, all three had dependent variables related to empathetic reactions to another persons’ plight. Furthermore, four studies reported that global avoidant style moderated the effects of primed style, diluting (weakening) the effect of the security prime weaker [25], [29] (a–c). Notably, none of these four studies used a repeated prime methodology.

### 3.4. Combining Subliminal and Supraliminal Priming Methods

Four studies used a combination of supraliminal and subliminal priming techniques. One reported that supraliminal and subliminal priming tasks similarly enhanced empathy and inhibited personal distress [17] (b) and another that they had comparable positive effects on depressive symptoms [26] (a). Conversely, one study found subliminal priming to be more effective than supraliminal priming [33] (a), while another found supraliminal priming to be more effective [31].

### 3.5. Quality Assessment

External validity. Samples were largely drawn from local universities and rewarded participation with course credits. Some studies used purpose sampling, based on characteristics of the population and research objectives [16,24], [26] (b), [27,31]. None reported the source population from which the sample was derived. Thus, it is not possible to determine the representativeness of samples. Furthermore, none provided information about whether the primes, staff or study materials were familiar to the participants, all of which could impact on the validity and reliability of the findings.

Internal validity. Most studies used reliable outcome measures (e.g., ECR, Profile of Mood States) and appropriate statistical tests (e.g., Analysis of Variance, *t*-Tests), and reported that the time between pre-measure, primes and follow-up were approximately consistent between participants. Nearly all studies using a repeated measure design controlled for order effects by counterbalancing their measures [29] (c), [33] (b–d)), apart from one study which omitted this information [31]. Most studies were carried out entirely under laboratory conditions which helped to standardise procedures, whilst other studies permitted participants to complete the procedure outside the laboratory [22] (b), [25], [26] (a,b), [27], [29] (a–c), [30]. Independent observer reports or task training were not reported in any of the studies, and only half included self-reports of prime task engagement or difficulty (e.g., the extent to which a participant felt engaged with the task or the ease with which a visualisation was achieved), or probed participants on the true purpose of the experiment afterwards [18], [29] (a–c), [33] (a–g). One study that did probe prime engagement [18] asked participants whether they were able to engage with the prime manipulation (e.g., rate on a scale of 1–7, the clarity of the visualisation during priming). Most studies stated that the participants were blind to their experimental condition, and a smaller number of studies used a double-blinded procedure [13,16], [33] (a–g). Filler questions or distraction tasks were used in approximately half of the studies to prevent the participant guessing the purpose of the research [13], [17] (a–e), [18], [22] (a,b), [24], [29] (a–c), [32] (a,b). Finally, only six studies conducted a manipulation check on felt security to determine whether the prime manipulation had successfully induced attachment security [13,18], [22] (a,b), [26] (b), [27].

Bias. Most studies suffered from a sampling bias as they recruited participants from local universities. Additionally, in most studies participants directly chose whether to be involved in the study, thus a self-selection bias occurred. Nearly all studies recruited participants from a single population, apart from two studies that used two different institutions or settings [13], [26] (b). Approximately half of the studies analysed group differences before administering the intervention in order to determine that the two groups were approximately equal in their baseline characteristics such as personality traits, socio-economic background, gender [13], [17] (a–e), [22] (a,b), [24], [26] (a,b), [27,31]. Data loss or exclusion was reported by half the studies [18], [22] (a,b), [24,25], [26] (a,b), [29] (a–c), [30], [32] (a,b), for reasons including missing responses, research errors, attrition, and exclusion due to failure to follow study instructions.

Power. Twenty studies reported power calculations to determine target sample size [17] (a–e), [22] (a,b), [25,27], [29] (a–c), [30], [33] (a–g). Sixteen studies were significantly powered to detect a large effect [17] (a–e), [22] (b), [25,27,30], [33] (a–g), whilst four studies were low on statistical power due to small samples sizes [22] (a), [29] (a–c).

### 3.6. Highest Rated Studies

The highest rated studies based on the quality checklist [34] were three [22] (b), [24,30]. The checklist produces a score from 0 to 27. The first [24] scored 23; notable strengths were their preliminary analyses to test for potential confounds, the use of multimodal assessment (e.g., physiological measurement instruments and self-report ratings) and the statistical procedures used to deal with missing data. The second [30] scored 22; strengths were the longitudinal design, high statistical power, and reporting of attrition analysis. The final study [22] (b) scored 22; strengths included their repeated prime methodology, attempts to blind participants to the outcomes of the experiment, and detailed reporting of statistical analyses, such as actual probability values and an explanation of how they dealt with outliers.

## 4. Discussion

The results of this systematic review suggest that attachment security priming effectively reduces negative affect and increases positive affect. Most studies report significant group differences (i.e., secure prime group compared to control groups) or within-participant differences post-intervention, with medium to large effect sizes. Studies used a range of methodologies and designs (e.g., within and between-participant experimental designs, subliminal and supraliminal priming techniques, one-time prime or repeated primes) and dependent and follow-up measures. Typically, questionnaires were used to measure attachment style, positive and negative affect, depression and anxiety, and felt security. Experimental designs generally compared an attachment security prime to a neutral, anxious or avoidant prime, although there was variability in priming procedures in terms of the frequency and time-lag. Overall, security priming in all its forms appeared effective in improving positive and reducing negative affect relative to control primes. This was the case across methods, designs, and dependent variables.

Comparable results were reported for both supraliminal and subliminal priming methods in reducing negative affect and increasing positive affect. That said, findings suggest that within each priming method some techniques may be better than others. The more effective supraliminal primes required participants to focus on (visualise and/or write about) their own attachment figures or experiences rather than attachment stimuli representing people or experiences outside their personal histories. Additionally, subliminal picture primes were more effective than word primes in increasing positive affect and decreasing negative affect. It should be noted, however, that only 6 studies used word primes and the only paper that used both subliminal words and pictures found them to be equally effective [33].

Notably, repeated priming studies suggest a cumulative positive effect of security priming over time. These designs, although more laborious, seem highly effective in keeping security elevated over time and can have long-lasting effects, up to a month after the last prime. Repeated priming may be particularly effective for insecure individuals, breaking down the defenses associated with insecurity over time to confer the benefits of security. Of the fifteen studies that examined whether the effects of the prime were moderated by global attachment style, approximately half reported this to be the case. Prime effects were moderated by both global avoidance and anxiety and mainly in one-shot prime studies. Moderation by global attachment anxiety was observed in one-shot prime studies involving empathy-evoking tasks. Such tasks may activate individuals’ global anxious attachment style because they induce feelings of distress [5]. Moderation by attachment avoidance was observed in one-shot prime studies involving diverse measures and dependent variables. These moderation effects further underline the value of repeated priming versus one-shot prime study designs. Repeated priming designs may be more effective than one-shot study designs in overcoming feelings of distress for individuals high in attachment anxiety and at penetrating the defensive mechanisms associated with global attachment avoidance.

### 4.1. Strengths of the Literature

It is a particular strength that most studies compared the experimental condition to an active control condition (where participants receive a similar intervention to the experimental group) as opposed to a passive control condition (in which no intervention is received). Passive controls can result in confounding variables and affect the validity of the study (e.g., amount of experimenter contact, expectancy effects and motivation [45], while active controls allow the possibility that participants may benefit from the alternative intervention [46].

Additional methodological strengths of studies reviewed were procedure randomisation, use of published self-report measures, blind condition assignment and matched group designs. One study applied a particularly strong methodology in a double-blind procedure [33] and another [22] reduced internal bias by blinding participants to the true study purpose. Within-subject design studies counterbalanced their trials to prevent order effects (although one did not report this information). Approximately two thirds of the studies examined the potential moderating effects of the prime by global attachment dimensions.

### 4.2. Limitations of the Literature

Methodological limitations included an over-reliance on university samples and female participants, effect sizes not being reported, limited recruitment of children or older adults, and studies measuring negative but not positive affect (and vice versa). In addition, most studies used self-report measures that are subject to recall, response bias (e.g., social desirability) and objectivity issues [47]. None of the studies distinguished between different types of avoidant attachment in priming avoidant style, potentially confounding fearful and dismissing attachment styles and their potential distinct effects on affect [48].

### 4.3. Future Research

Future research directions are suggested by the review. Firstly, future research might include post-intervention follow-up data collection over a longer term than has been done to date. This would allow researchers to determine whether security primes have long-term effects on outcomes and precisely how long these effects last. A general criticism of priming studies is that they produce short-term effects [49,50]. Repeated priming, however, seems to result in relatively longer lasting effects [51]. Indeed, the repeated priming studies reviewed here support this notion, showing that the positive priming effects can be maintained for several days post-prime. A next step for research is to use a repeated priming methodology with affect as the principle dependent variable and to include post-prime follow-ups in excess of 1-week post-last prime, (perhaps 1 month and 6 months post-last prime).

Secondly, attempts should be made to triangulate self-report questionnaires with assessment of the individuals’ behavioural and physiological measures in security priming studies. This would strengthen research findings [52] and point to physiological and behavioural benefits of attachment security priming. It would also allow the examination of potential physiological mediators or moderators of the effects of security of attachment on affect. It may be the case, for example, that the benefits of security priming for psychological wellbeing and affect are mediated by physiological effects.

Thirdly, given the mixed findings regarding the moderation of security priming by global attachment dimensions, well-powered longitudinal research is needed. If global attachment dimensions reliably moderate the effect of security priming this will have implications for study methods. For example, a longitudinal priming study could examine whether repeated security priming designs are more effective at overcoming the defensive emotion regulation strategies of individuals that are high in global avoidance than one-shot priming studies.

Fourthly, given the medium to large effect sizes that were generally reported, a meta-analysis of this literature, focusing on particular types of primes, for example, may be a fruitful future direction. Unfortunately, given the range of priming techniques and measures used across our reviewed studies, the aggregation of effect sizes would be minimally informative.

Finally, our results have clinical implications for people generally, but specifically too for young people and children. Only two studies in the current review were conducted with children and young people below 18 years old and this is an important direction for research. In 2017, the UK Office of National Statistics reporting that that one in eight children and young people (aged between 5 to 19 years old.) had a mental disorder (mental disorders were identified according to International Classification of Diseases (ICD-10) to standardise diagnostic criteria. Mental Disorders were defined by symptoms that caused significant distress to children and young people or impaired their functioning.), and one in twelve had an emotional disorder such as anxiety or depression [53]. Future research should explore the impact of security priming with samples of children and young people with the aims of examining how to improve emotional wellbeing and of designing therapeutic and clinical interventions.

## 5. Conclusions

This systematic review represents the first thorough quality assessment of the literature on attachment security priming and affect. The findings reported herein will be useful to attachment researchers in designing future studies. Notably, clear inclusion and exclusion criteria helped to minimise the possibility of study selection bias.

While a small number of studies failed to observe priming influences on affect, overall, our findings show that security priming effectively reduces negative affect, increases positive affect and has beneficial effects across a diverse set of outcomes. The review points to the value of repeated security priming study designs, which may be more effective at breaking down the defenses associated with attachment insecurity, and the use of both subliminal and supraliminal security primes.

## Figures and Tables

**Figure 1 ijerph-17-00968-f001:**
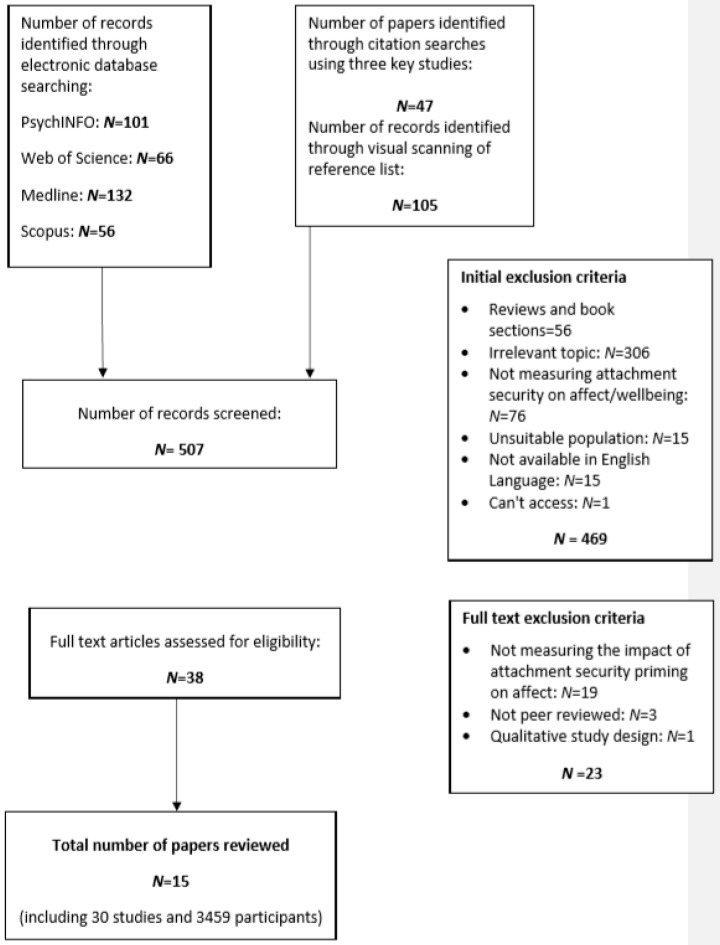
Chart of Search and Retrieval.

**Table 1 ijerph-17-00968-t001:** Summary of Studies Included in the Systematic Review.

Authors & Date	Country	Population	Design & Prime Type	ASP Intervention	Main Affect Findings	Interaction with Attachment	Effect Size
Bryant & Chan (2017) [25]	UK	75 university students Age: *M* = 19.25 56 females, 19 males	Experimental, Between-subject Supraliminal priming Attachment security prime vs. control (positive prime)	Mental imagery task 3 min duration	Participants with low avoidant attachment style who received the secure prime reported less distress than those who received the control prime. This pattern was not found for participants with a high avoidant attachment style.	Attachment style moderated the effects of the prime as the findings showed that individuals with a high avoidant attachment style were not impacted by the prime.	Personal distress *η*^2^ = 0.18; *η*^2^ = 0.19 (Large) Positive affect *η*^2^ = 0.11 (Medium)
Carnelley & Rowe (2007) [13]	UK	64 university students Age: 18–55 (*M* = 21.18) 46 females, 18 males	Experimental, Between-subject Supraliminal priming Attachment security prime vs. control (neutral prime)	Mental imagery and written task 10 min duration Primed on 3 occasions across 3 days	Repeated priming of attachment security resulted in more positive self-views and less attachment anxiety at Time 5 compared to Time 1. Those primed with neutral primes showed no change with time.	Attachment dimensions did not moderate the effects of the prime for self-views.	Positive self-views *η*p^2^ = 0.25 (Large)
Carnelley, Otway, & Rowe (2016) [22] **(a)** Study 1	UK	144 university students Age: 18–50 (*M* = 20.1) 127 females, 17 males	Experimental, Between-subject Supraliminal priming Attachment security prime vs. anxious prime; avoidant prime; control (neutral prime)	Mental imagery and written task 10 min duration	Anxious-primed participants reported higher depressed mood than secure-primed participants. Avoidant-primed and anxious-primed participants reported higher anxious mood compared to secure-primed participants. Secure-primed participants did not report significantly lower anxious or depressed mood than neutral-primed participants.	Attachment dimensions did not moderate the effects of the prime.	Depressed mood *η*^2^ = 0.21 (Large) Anxious mood *η*^2^ = 0.22 (Large)
Carnelley, Otway, & Rowe (2016) [22] **(b)** Study 2	UK	81 university students Age: 18–33 (*M* = 20.32) 70 females, 11 males	Experimental, Between-subject Supraliminal priming Attachment security prime vs. control (neutral prime)	Mental imagery and written task 10 min duration for initial prime 3 min duration for 3 subsequent primes	Secure primed participants reported lower anxious mood post-prime and one day later compared with neutral-primed participants. Secured-primed participants reported marginally lower depressed mood post-prime and one day later compared to neutral-primed participants.	Attachment dimensions did not moderate the effects of the prime.	Anxious mood 11% of the variance (Medium)
McGuire, Gillath, Jackson, & Ingram (2018) [26] **(a)** Study 1	US	125 college students Age: 18–47 (*M* = 19.6) 71 females, 54 males	Experimental, Mixed methods Supraliminal and Subliminal priming Attachment security prime vs. control (neutral prime)	Computerised lexical decision task which included rapid subliminal presentation of prime words (prime 24 milliseconds) Mental imagery and written task (5 min)	Participants exposed to the security primes reported a greater decrease in depressive symptoms compared to participants exposed to neutral primes.	Attachment dimensions did not moderate the effects of the prime, although attachment anxiety was significantly associated with depressive symptoms.	N/A
McGuire, Gillath, Jackson, & Ingram (2018) [26] **(b)** Study 2	US	69 adolescents from high school/youth centre. Age: 13–19 (*M* = 15.6) 39 females, 30 males	Experimental, Between-subject Supraliminal priming Attachment security prime vs. control (neutral prime)	Mental imagery and written task (5 min) Word search task Mental imagery task (3 min) Picture writing task (2 min) Sentence memorisation task (3 min)	Adolescents who were repeatedly exposed over two weeks to security primes showed lower depression symptoms than participants exposed to neutral primes.	Attachment dimensions did not moderate the effects of the prime, although attachment anxiety was significantly associated with depressive symptoms.	N/A
Stupica, Woodhouse, Brett, & Cassidy (2017) [24]	US	90 school children Age: 6–7 (*M* = 6.95) 42 females, 48 males	Experimental, Between-subject Subliminal priming Attachment security prime vs. controls (happy prime; neutral prime)	Computer picture presentation which included subliminal picture priming Primes presented for 24 milliseconds	Secure priming decreased physiological responses (electrodermal activity, vagal augmentation, fearful facial expressions) to threat compared to control conditions. There were no priming effects associated with children’s self-reported fear.	Attachment dimensions did not moderate the effects of the prime, although securely attached children had lower physiological responses to fear. There were no attachment effects associated with children’s self-reported fear.	Electrodermal activity *d* = 1.79; *d* = 1.84 (Large) Respiratory sinus arrhythmia *d* = 1.95; *d* = 3.40 (Large) Fearful facial expressions *d* = 15.00; *d* = 15.92 (Large)
Rowe & Carnelley (2003) [18]	UK	160 university students Age: 17–42 (*M* = 20.5) 121 females, 39 males	Experimental, Between-subject Supraliminal priming Attachment security prime vs. avoidant prime and anxious prime	Mental imagery and written task 10 min duration	Primed secures reported more positive affect and less negative affect compared to the other primed attachment style groups	Attachment dimensions did not moderate the effects of the prime.	N/A
Mikulincer, Gillath, Halevy, Avihou, Avidan, & Eshkoli (2001) [17] **(a)** Study 1	Israel	69 university students. Age: 20–40 (*Mdn* = 24) 44 females, 25 females	Experimental, Between-subject Supraliminal priming Attachment security prime vs. controls (positive-affect prime; neutral prime)	Reading an interpersonal script related to attachment security Duration unspecified	Priming attachment security and positive-affect led to lower ratings of personal distress compared to the neutral priming. There was not a significant difference in personal distress between secure prime group and positive-affect group.	Attachment anxiety had a significant unique effect on personal distress; the higher the attachment anxiety the higher the reported distress. Main effect of avoidance and all the interactions were not significant.	Personal distress 28% of variance (Large)
Mikulincer, Gillath, Halevy, Avihou, Avidan, & Eshkoli (2001) [17] **(b)** Study 2	Israel	60 university students Age: 17–39 (*Mdn* = 24) 31 females, 29 males	Experimental, Between-subject Supraliminal and Subliminal priming Attachment security prime vs. controls (positive-affect prime; neutral prime)	1. Subliminal exposure of prime images. Duration unspecified 2. Reading a distressing interpersonal script. Duration unspecified	Attachment security priming and positive-affect priming led to lower passive identification (sorrow-related emotions) ratings compared to neutral priming. There was not a significant difference in personal distress between secure prime group and positive-affect group.	Attachment anxiety had a significant unique effect on personal distress; the higher the attachment anxiety the higher the reported distress. Main effect of avoidance and all the interactions were not significant.	Passive identification 20% of variance (Large)
Mikulincer, Gillath, Halevy, Avihou, Avidan, & Eshkoli (2001) [17] **(c)** Study 3	Israel	60 university students Age: 19–30 (*Mdn* = 23) 34 females, 26 males	Experimental, Between-subject Subliminal priming Attachment security prime vs. controls (positive -affect prime; neutral prime)	Computerised lexical decision task which included rapid subliminal presentation of prime words.	Attachment security priming led to lower ratings of personal distress compared to neutral priming. Security priming also led to lower personal distress than positive-affect priming.	Attachment anxiety had a significant unique effect on personal distress; the higher the attachment anxiety the higher the reported distress. Main effect of avoidance and all the interactions were not significant.	Personal distress 33% of variance (Large)
Mikulincer, Gillath, Halevy, Avihou, Avidan, & Eshkoli (2001) [17] **(d)** Study 4	Israel	72 university students Age: 20–37 (*Mdn*= 24) 37 females, 35 males	Experimental, Between-subject Subliminal priming Attachment security prime vs. controls (positive-affect prime; neutral prime)	Computerised autobiographical memory task which included subliminal picture priming. Duration unspecified.	There was not a significant difference in accessibility to personal distress memories between secure prime, positive-affect prime and neutral prime groups.	Attachment anxiety had a significant unique effect on personal distress; the higher the attachment anxiety, the higher the accessibility of personal distress memories. Main effect of avoidance and all the interactions were not significant.	Personal distress 10% 0f variance (Medium)
Mikulincer, Gillath, Halevy, Avihou, Avidan, & Eshkoli (2001) [17] **(e)** Study 5	Israel	150 university students Age: 18–27 (*Mdn* = 23) 66 females, 84 males	Experimental, Between-subject Supraliminal priming Attachment security prime vs. attachment anxiety, attachment avoidance, positive-affect, neutral	Two mental imagery tasks 2 min durations	The priming of attachment security led to lower personal distress than the priming of attachment anxiety. Priming attachment anxiety led to higher personal distress than the priming of avoidance.	Attachment anxiety had a significant unique effect on personal distress; the higher the attachment anxiety the higher the reported distress. Main effect of avoidance and all the interactions were not significant.	Personal distress 34% of variance (Large)
Liao, Wang, Zhang, Zhou, & Xiangping (2017) [16]	China	105 university students Age: 17–27 (*M* = 20.3) 70 females, 35 males	Quasi-experimental, Between-subject Supraliminal priming Attachment security prime (no control)	Mental imagery and written task 10 min duration Primed once	Individuals with dependent depression experienced greater positive affect after priming. There was no significant change in positive affect after priming for individuals with self-critical depression.	Not tested	N/A
Carnelley, Bejinaru, Otway, & Baldwin (2018) [27]	UK	48 adults with depressive disorder Age: 18–76 (*M* = 50.9) 29 females, 19 males	Experimental, Between-subject Supraliminal priming Attachment security prime vs. control (neutral prime)	Mental imagery task and written task 10 min duration for initial prime 3 min duration for subsequent primes on 3 consecutive days	Secure priming had a greater impact on reducing symptoms of anxiety and depression in comparison to the control prime, though the differences were only significant at Time 4 (third and last text prime).	Not tested on anxiety and depression.	Depression *η*p^2^ = 0.10 (Medium) Anxiety *η*p^2^ = 0.13 (Medium)
Cassidy, Shaver, Mikulincer, & Lavy (2009) [28]	US	70 university students 17–25 (*Mdn* = 19) 51 females, 19 males	Experimental, Between-subject Subliminal priming Attachment security prime vs. control (neutral prime)	Computerised cognitive categorization task which included subliminal priming of words. Prime presented for 22 milliseconds	Attachment security priming influenced the participants responses to psychological pain (operationalised in terms of hurt feelings) in different ways depending on global attachment style (see interaction with attachment). Overall, security priming was able to reduce some maladaptive responses to psychological pain (e.g., over-engagement and under-engagement with negative emotions) within insecurely-attached individuals.	Avoidant attachment was associated with a tendency to dismiss hurtful events, inhibit expressions of distress and react hostilely in the neutral prime condition, and it was associated with greater openness to pain in the security prime condition. Attachment anxiety was associated with more intense feelings of rejection, more crying and more negative emotions in the neutral prime condition, but interactions were generally non-significant within the security prime condition.	Rejected feelings 23.5% of variance (Large) Negative emotions 32.1% of variance (Large) Positive emotions 5.8% of variance (Small)
Selcuk, Zayas, Günaydin, Hazan, & Kross (2012) [29] **(a)** Study 1	US	123 university students Age: *M* = 20 105 females, 18 males	Experimental, Mixed design Supraliminal priming Attachment security prime vs. control (acquaintance prime)	Mental imagery task 20 s at a time	Priming participants with attachment security after recalling an upsetting memory led to significantly lower negative affect in comparison to neutral priming (recovery hypothesis). Priming participants before an upsetting memory recall did not result in significant differences between the security prime and neutral prime conditions (buffering hypothesis).	Individuals high on attachment avoidance showed less affective recovery as a result of priming the mental representation of an attachment figure. Higher attachment anxiety towards one’s mother was associated with smaller recovery effects (although not statistically significant).	Negative affect *η*p^2^ = 0.22 (Large)
Selcuk, Zayas, Günaydin, Hazan, & Kross (2012) [29] **(b)** Study 2	US	139 university students Age: *M* = 20 105 females, 34 males	Experimental, Mixed design Supraliminal priming Attachment security prime vs. control (stranger prime)	Exposure to photograph of mother 90 s at a time	Priming participants with attachment security after recalling an upsetting memory led to significantly lower negative affect in comparison to neutral priming (recovery hypothesis). Priming participants before an upsetting memory recall did not result in significant differences between the security prime and neutral prime conditions (stranger; buffering hypothesis).	Individuals high in attachment avoidance showed less affective recovery as a result of priming the attachment figure. Higher attachment anxiety towards one’s mother was associated with smaller recovery effects (although not statistically significant).	Negative affect *η*p^2^ = 0.37 (Large)
Selcuk, Zayas, Günaydin, Hazan, & Kross (2012) [29] **(c)** Study 3	US	57 members of heterosexual romantic couples Age: *M* = 21 29 females,28 males	Experimental, Within-subject Supraliminal priming Attachment security prime vs. control (stranger prime)	Exposure to photograph of romantic partner 90 s at a time	Priming participants with attachment security after recalling an upsetting memory led to significantly lower negative affect in comparison to neutral priming (recovery hypothesis). After recalling the upsetting memory, participants in the secure prime condition showed lower negative thinking compared to control condition.	Individuals high in attachment avoidance showed less affective recovery as a result of priming the attachment figure.	Negative affect *η*p^2^ = 1.21 (Large)
Hudson & Fraley (2018) [30]	US	133 university students Age: *M* = 20.15 92 females, 41 males	Experimental. Between-subject Supraliminal priming Attachment security prime vs. anxious prime; control (no prime)	Mental imagery and written tasks Duration unspecified Primed once a week for over 16 weeks	There was no significant improvement in emotional wellbeing over the course of 4 months for either the security prime condition or the attachment anxiety prime condition.	Attachment dimensions did not moderate the effect of the prime on wellbeing.	N/A
Canterberry & Gillath (2012) [31]	US	30 men and women Age: 18–24 (*M* = 21.4) 15 females, 15 males	Experimental, Within-subject Supraliminal and Subliminal priming Attachment security prime vs. insecure attachment prime vs. control (neutral prime)	Event-related computerised priming task which included being primed with words. Implicit prime presented for 2 milliseconds Explicit prime presented for 500 milliseconds	Supraliminal security priming led to higher liking ratings for the images compared to insecurity or neutral primes. No significant differences were found for subliminal priming. Attachment security priming activated unique brain areas related to affect.	Attachment dimensions did not moderate the effects of the prime.	Positive affect *η*p^2^ = 0.22 (Large)
Dutton, Lane, Koren, & Bartholomew (2016) [32] **(a)** Study 1	Canada	686 university students Age: 18–59 (*M* = 20.4) 505 females, 181 males	Experimental, Between-subject Subliminal priming Attachment security prime vs. controls (distraction prime; no prime)	Exposure to prime images after listening to audio recordings of interpersonal conflict. Duration unspecified	The attachment security priming group reported significantly lower anger and anxiety scores compared to the control groups.	Not tested	N/A
Dutton, Lane, Koren, & Bartholomew (2016) [32] **(b)** Study 2	US	278 internet sample Age: 18–16 (*M* = 34.6) 163 females, 115 males	Experimental, Between-subject Subliminal priming Attachment security prime vs. controls (smiling man; cold mother; no prime)	Exposure to prime images after listening to audio recordings of interpersonal conflict, with two additional controls. Duration unspecified	The attachment security priming group reported significantly lower anger and anxiety scores compared to the control groups.	Not tested.	N/A
Mikuliner, Hirschberger, Nachmias, & Gillath (2001) [33] **(a)** Study 1	Israel	106 university students Age: 21–31 (*Mdn* = 24) 79 females, 27 males	Experimental, Between-subject Subliminal and Supraliminal priming Attachment security prime (baby and mother) vs. controls (positive affect prime; neutral prime; control prime)	Rated Chinese ideographs whilst being primed by pictures. Supraliminal prime presented for 500 milliseconds Subliminal prime presented for 10 milliseconds	In the subliminal trials, attachment security priming and positive-affect primes led to higher liking ratings compared to neutral or no primes. No significant difference was found between security primes and positive-affect primes. There were no significant differences between prime groups in the supraliminal trials.	Attachment dimensions did not moderate the effect of the primes.	Positive affect *η*p^2^ = 0.07 (Medium)
Mikuliner, Hirschberger, Nachmias, & Gillath (2001) [33] **(b)** Study 2	Israel	45 university students Age: 19–32 (*Mdn* = 22) 33 females, 12 males	Experimental, Within-subject Subliminal priming Attachment security primes (baby and mother; baby prime; mother prime) vs. controls (positive-affect prime; neutral prime)	Rated Chinese ideographs whilst being subliminally primed by pictures. Subliminal prime presented for 10 milliseconds	‘Baby and mother’ security primes and positive affect primes led to higher liking ratings compared to neutral primes, baby primes and mother primes.	Attachment dimensions did not moderate the effect of the primes.	Positive affect *η*p^2^ = 0.15 (Large)
Mikuliner, Hirschberger, Nachmias, & Gillath (2001) [33] **(c)** Study 3	Israel	40 university students Age: 20–32 (*Mdn* = 23) 29 females, 11 males	Experimental, Within-subject Subliminal priming Attachment security prime (baby and mother; young couple prime; old couple) vs. controls (positive prime; neutral prime)	Rated Chinese ideographs whilst being subliminally primed by pictures. Subliminal prime presented for 10 milliseconds	All security primes and positive affect prime led to higher liking ratings compared to neutral prime. No significant differences were found between any of the attachment security primes.	Attachment dimensions did not moderate the effect of the primes.	Positive affect *η*p^2^ = 0.18 (Large)
Mikuliner, Hirschberger, Nachmias, & Gillath (2001) [33] **(d)** Study 4	Israel	42 university students Age: 19–35 (*Mdn* = 22) 31 females, 11 males	Experimental, Within-subject Subliminal priming Attachment security prime (attachment figure) vs. controls (close person; known person; unknown person)	Rated Chinese ideographs whilst being subliminally primed by names. Subliminal prime presented for 10 milliseconds	Attachment security priming led to higher liking ratings compared to all control primes. No significant difference was found between control primes.	Attachment dimensions did not moderate the effect of the primes.	Positive affect *η*p^2^ = 0.16 (Large)
Mikuliner, Hirschberger, Nachmias, & Gillath (2001) [33] **(e)** Study 5	Israel	150 university students Age: 20–39 (*Mdn* = 24) 94 females, 56 males	Experimental, Between-subject Subliminal priming Attachment security prime vs. controls (positive affect; neutral; no prime)	Rated Chinese ideographs whilst being subliminally primed by words and picture. Two different contexts were induced: neutral and threat conditions. Subliminal prime presented for 10 milliseconds	In the neutral context, attachment security primes and positive affect primes led to higher liking ratings than neutral primes or no primes. In the threat context, attachment security priming led to higher liking ratings than all control primes (including positive-affect).	Attachment dimensions did not moderate the effect of the primes.	Positive affect *η*p^2^ = 0.14 (Large)
Mikuliner, Hirschberger, Nachmias, & Gillath (2001) [33] **(f)** Study 6	Israel	88 university students Age: 20–35 (*Mdn* = 24) 53 females, 35 males	Experimental, Between-subject Subliminal priming Attachment security prime vs. controls (positive affect; neutral; no prime)	Rated Chinese ideographs after the subliminal presentation of picture primes. Two different contexts were induced: no feedback condition and failure condition. Subliminal prime presented for 10 milliseconds	In the no feedback condition, attachment security and positive affect primes led to higher liking ratings that neutral and no picture primes. In the failure condition, attachment security priming led to higher liking ratings compared to all the control primes (including positive-affect).	There was a significant main effect for attachment anxiety; individuals with high levels of attachment anxiety reported lower liking ratings than individuals who scored low in attachment anxiety. The main effect for avoidance and the remaining interactions were not significant.	Positive affect *η*p^2^ = 0.13 (Large)
Mikuliner, Hirschberger, Nachmias, & Gillath (2001) [33] **(g)** Study 7	Israel	100 university students Age: 20–35 (*Mdn* = 24) 72 females, 28 males	Experimental, Between-subject Subliminal priming Attachment security prime vs. controls (positive affect; neutral; no prime)	Rated Chinese ideographs after the subliminal presentation of picture primes. Two different contexts were induced: neutral and visualisation of a separation. 1. Mental imagery and written task 3 min duration 2. Rated Chinese ideographs after the subliminal presentation of picture primes Subliminal prime presented for 10 milliseconds	In the neutral condition, attachment security prime and positive affect prime led to higher liking ratings than neutral or no picture primes. In the separation episode, attachment security priming led to higher liking ratings compared to all control primes (including positive-affect).	Attachment dimensions moderated the effect of the prime following the visualisation of a separation episode. Attachment dimensions did not moderate the effects of the prime in the neutral context.	Positive affect *η*p^2^ = 0.18 (Large)

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
