# Peer review of "The Effectiveness of Attachment Security Priming in Improving Positive Affect and Reducing Negative Affect: A Systematic Review"

_ijerph, 2020, doi:10.3390/ijerph17030968_

Round 1
Reviewer 1 Report
The authors made a systematic review, which thoroughly reviewed the empirical researches on the topic of the effectiveness of attachment securing priming in improving positive affect and reducing negative affect. Strengths of the review include the well written of this manuscript and a detailed comparison of multiple methods of reviewed literature, which might have implications for future researches. I think this review present a contribution to the filed and could be accepted after minor revision. There are two suggestions for authors: 1.The theoretical rational linking attachment security to positive affect and negative affect of the present review are not very strong. It would be helpful for authors to demonstrate the theoretical rational behind the links of the present review. 2. In the discussion section, the authors mainly summarized the existing literature but detailed little about the innovation and contribution of this review, It would also be helpful for authors to demonstrate the innovation and contribution of the present review to the existing literature.
Author Response
Response to Reviewer 1
We thank Reviewer 1 for their positive and thoughtful consideration of the manuscript. Below we address each point made by Reviewer 1. Our responses are in italics.
The authors made a systematic review, which thoroughly reviewed the empirical researches on the topic of the effectiveness of attachment securing priming in improving positive affect and reducing negative affect. Strengths of the review include the well written of this manuscript and a detailed comparison of multiple methods of reviewed literature, which might have implications for future researches. I think this review present a contribution to the field and could be accepted after minor revision.
We thank the Reviewer for their kind words.
There are two suggestions for authors:
The theoretical rational linking attachment security to positive affect and negative affect of the present review are not very strong. It would be helpful for authors to demonstrate the theoretical rational behind the links of the present review.This is an excellent point. We have addressed it by specifying the importance of attachment style as a predictor of affect regulation (see line 37-39). The manuscript now reads: “Attachment styles are important predictors of the way individuals regulate affect. Individuals learn through relationships how and when to attend to their own stress and distress (Mikulincer & Shaver, 2016)”. To further develop the theoretical rationale we now also say (from line 43): Individuals who are high in attachment anxiety find it hard to regulate their emotions. They use hyperactivation emotion regulation strategies, that is, they are hypervigilant for signs of rejection and have relationships characterised by emotional turbulence. Individuals high in attachment avoidance use deactivating emotion regulation strategies, that is, they ignore or deny emotional threats and tend toward compulsive self-reliance.
In the discussion section, the authors mainly summarized the existing literature but detailed little about the innovation and contribution of this review. It would also be helpful for authors to demonstrate the innovation and contribution of the present review to the existing literature.Thank you to Reviewer 1 for pointing this out. We have now re-written the ‘Conclusions and Future Directions’ subsection to include this. The first paragraph of this section now reads: “This systematic review represents the first thorough quality assessment of the literature on attachment security priming and affect. The findings reported herein will be useful to attachment researchers in designing future studies”.
Reviewer 2 Report
This is an excellent paper on a very important area (and under researched in my view) of attachment. The authors have developed a coherent and extensive methodology for investigating therapuetic interventions to help support people with attachment issues. This survey will have long range applications in help the field to develop. The research takes us beyond attachment findings by showing how intervention can be beneficial. There is no reference to origin of attachment difficulties, but this is for other papers. This is an excellent survey paper.
More research on attachment and therapuetic interventions.
Author Response
Response to Reviewer 2
We thank Reviewer 2 for their very positive and thoughtful consideration of the manuscript. Our response to the issue raised by Reviewer 2 is in italicised font below.
This is an excellent paper on a very important area (and under researched in my view) of attachment. The authors have developed a coherent and extensive methodology for investigating therapeutic interventions to help support people with attachment issues. This survey will have long range applications in help the field to develop. The research takes us beyond attachment findings by showing how intervention can be beneficial. There is no reference to origin of attachment difficulties, but this is for other papers. This is an excellent survey paper.
Many thanks to Reviewer 2 for their kind words.
More research on attachment and therapeutic interventions.Thank you for pointing out that we should stress that the review could inform research on attachment in the context of therapeutic interventions. We have incorporated this point at the end of the manuscript. The final sentence now reads: “Future research should explore the impact of security priming with samples of children and young people with the aims of examining how to improve emotional wellbeing and of designing therapeutic and clinical interventions”.